# Cell Classification from Multi-Focus Images by Deep Neural Network

Ken'ichi Morooka[1], Jun Ohta[1], Shoko Miyauchi[1]
Ryo Kurazume[1], and Eiji Ohno[2]

[1] Graduate School of Information Science and Electrical Engineering
Kyushu University, Fukuoka 819-0395, Japan `morooka@ait.kyushu-u.ac.jp`
[2] Faculty of Health Sciences, Kyoto Tachibana University, Kyoto 607-8175, Japan

**Abstract.** This paper presents a new method for automatically classifying normal and abnormal cells from multi-focus images by deep neural networks (DNNs). There are the methods for classifying cells by using their images captured at a specific focus point. However, the use of such single-focus images sometimes makes it difficult to identify abnormal cells, especially, pre- cancerous cells with similar visual attributes with normal cells. To solve this problem, we propose a new method for automatically classifying normal and abnormal cells from multi-focus images by combining deep neural networks (DNNs) with different architectures. From the experimental results, our proposed system can classify cells with high performance.

**Keywords:** Cytodiagnosis · Multi-focus images · Deep Neural Network

## 1 Introduction

Cytodiagnosis is an effective and reliable method of early cancer diagnosis because of its less invasive natures. In the diagnosis, cytotechnologists observe a tissue sample taken out from human body, and find abnormal cells including precancerous and cancer cells from the sample. The detection of the abnormal cells is based on the difference of visual attributes between the normal and abnormal cells.

Generally, one sample includes tens of thousands of cells. Among them, the number of cancer cells is much smaller than that of normal cells. Moreover, in the case of cervical cancer screening in Japan, only 120 of every 10,000 people may carry cancer cells, and 7 of them will be diagnosed as suffering from cancer. Owing to these, the detection of cancer cells is a hard and time-consuming task.

Recently, instead of the sample, whole slide images (WSIs) have become a common method for not only cancer screening but also another clinical applications [1]. WSI is a high resolution digital image with gigapixels acquired by scanning the enter sample and varying a focus point. WSI has the potential to improve the accuracy and efficiency of the cancer screening including web-based remote diagnosis.

One application using WSIs is an automatic screening systems [2] that detect cancer cells automatically from pathological images. Moreover,there have developed fundamental techniques for the screening systems [2–8]. Especially, since deep learning methods using convolution neural networks(CNNs) have gained great popularity in medical image analysis, CNN-based techniques [3–8] for the screening systems have been proposed including the detection and semantic segmentation of the nuclei of cells in a given sample [3, 4]. Moreover, there are several CNN-based approaches for classifying cells in a cropped patch from WSIs [5–8]. Bora et al. [8] constructed the system for detecting cervical dysplasia by integrating three different classifiers: least square support vector machine, multilayer perceptron and random forest.

The conventional automatic screening systems use only the images of a sample captured at a specific focus point. However, the use of such single-focus images sometimes makes it difficult to identify abnormal cells, especially, precancerous cells with similar visual attributes with normal cells. Therefore, the conventional automatic screening systems can find cancer cells stably while detecting pre-cancerous cells with low accuracy. Especially, the latter detection is essential for early treatment of cancers.

Here, in the diagnosis, cytotechnologists observe the sample by varying a focus setting to obtain more information about the cells. In other words, to classify the cells, cytotechnologists uses multi-focus images of the sample acquired at different focus. On the contrary, considering the WSI acquisition process, WSI is also regarded as a sequence of multi-focus images of the sample. Therefore, the use of multi-focus cell images has the potential to improve the accuracy of detecting pre-cancerous cells. However, there are few methods using multi-focus images to detect abnormal cells.

In this paper, we propose a new method for automatically classifying normal and abnormal cells from multi-focus images by deep neural networks (DNNs). The advantage of combining multiple classifiers is that even though the accuracy of a classier may be low, the combination of multiple weak classifiers by using ensemble techniques results in a stable system of classifying cells. This is the reason why the combination of different classifiers are introduced to construct our cell classification system.

Similar with our method, Bora et al. [8] introduce an ensemble learning to construct the cell classification system by integrating several kinds of classifiers and hand-crafted visual features including shape and texture. On the contrary, our system is end-to-end system trainable with no manual definition of features used in the classification.

## 2   Material

Our dataset consists of the images of 24,540 cells. All the images are obtained by the samples taken out from women undergoing cervical cancer screening. For each sample, a digital slide scanner (Hamamatsu Photonics: Nanozoomer-XR) is used to acquire a whole slide image of the sample. The WSI contains 5 multi-

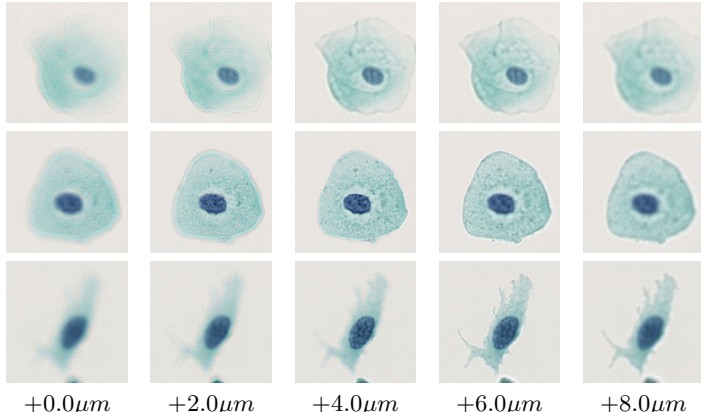

$+0.0\mu m$     $+2.0\mu m$     $+4.0\mu m$     $+6.0\mu m$     $+8.0\mu m$

Fig. 1: Examples of multi-focus images of cells: (top) NILM, (middle) LSIL, (bottom) HSIL.

focus images of the sample at intervals of $2.0[\mu m]$ focus length. Each WSI has $75,000 \times 75,000$ [pixel] while the spatial resolution of each image in the WSI is $0.23\mu$m/pixel. The multi-focus images of a target cell is extracted automatically from the WSI. The size of each cell image is $128 \times 128$ [pixel]. Accordingly, each cell data of the dataset has $128 \times 128 \times 5$ images.

Experienced cytotechnologists manually classify and label the cell images according to the Bethesda system. There are six categories of the cells in the Bethesda system. Our method uses three categories of the cells: negative for intraepithelial lesion or malignancy (NILM), low-grade squamous intraepithelial lesion (LSIL), and high grade squamous intraepithelial lesion (HSIL). Among them, LSIL and HSIL cells are regraded as cancer cells while NILM cells are used as normal cells. The dataset contains 24,000 NILM, 125 LSIL and 244 HSIL cells. The rest 171 cells are definitely cancer cells, but difficult for the experienced cytotechnologists to judge their cancer stage. Fig. 1 shows the example images of NILM (top in Fig. 1), LSIL (middle), and HSIL (bottom), respectively. In Fig.1, the values $+0.0, +2.0, \cdots, +8.0[\mu m]$ are related to the focus length. When the value of the focus length is close to 0, the sample is close to a lens of the scanner.

As mentioned above, in our dataset, the number of the abnormal cells is extremely smaller than that of the normal cells. To increase the number of the abnormal cells, we perform data augmentation as follows: rotating, translating, cropping, a mirror flip of the up-down and left-right directions, and changing brightness. Also, our data augmentation is applied to normal cell images to increase the variation of the normal cell images.

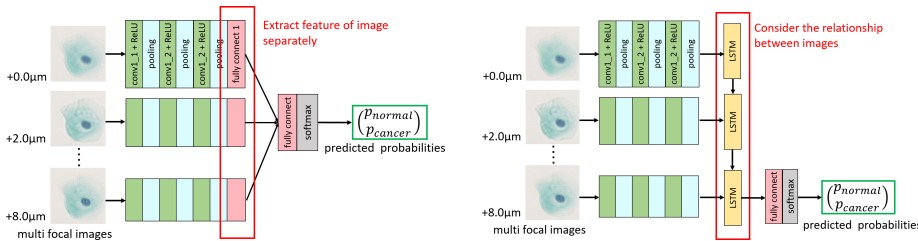

Fig. 2: Architectures of two new proposed DNNs using multi-focus cell images: (left) DNN-FC; (right) DNN-LSTM.

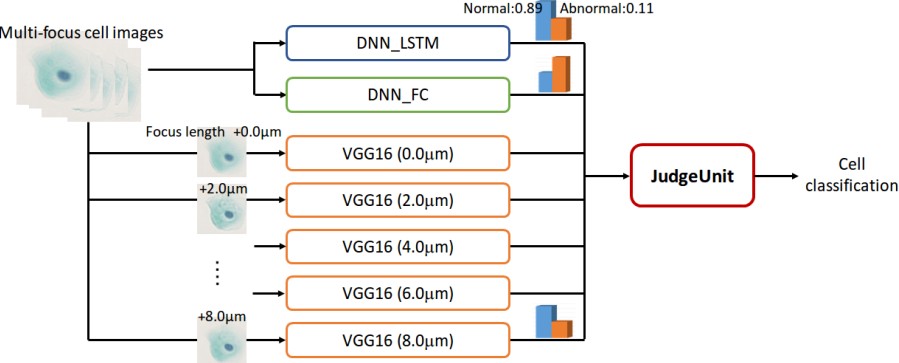

Fig. 3: Overview of our cell classification system.

## 3  Method

To detect abnormal cells stably and accurately, our cell classification system is generated by combining multiple DNNs, called classifiers, with different architectures. Fig. 3 shows the overview of our cell classification system. In the following, we explain the definitions of the classifiers forming our cell classification system, and the integration of the classifiers.

### 3.1  Classifiers

There are mainly two types of our classifiers according to their input data. The first type classifier uses one cell image as its input data. In our method, the pretrained VGG16[11] by ImageNet is used as the first type classifier. Since each cell data is the set of five multi-focus images of the cell, there are five VGG16s using cell images acquired at its corresponding focus point.

On the contrary, the second type classifier uses as its input the multi-focus images of the cell. Since there is few methods using multi-focus images as men-

tioned in Sec. 1, we construct two new DNNs: DNN-FC and DNN-LSTM (Fig. 2).

In both DNN-FC and DNN-LSTM, the first phase is to extract useful feature maps from each cell image by convolutional neural networks (CNNs). Therefore, there are five CNNs in DNN-FC and DNN-LSTM. Each CNN is composed of three convolutional layers, three max pooling layers and one fully connected layer.

After the feature extraction process, DNN-FC obtains the concatenated feature by the five feature maps. The concatenated feature is used as the input of a fully connected network (FCN) with a softmax output layer. On the contrary, in DNN-LSTM, the five feature maps are inputted to Long short-term memory (LSTM) in the order of their focus points. The aim of using LSTM is to obtain the feature considering the relationships between the images acquired at successive focus points. Similar with DNN-FC, the output of LSTM is used as the input of a fully connected network (FCN) with a softmax output layer.

The output of all the classifiers is a pair of the possibilities of normal and abnormal cells.

### 3.2   System construction

Given a cell data composed of multi-focus images, each classifier in our system identifies the cell in the input image separately. Moreover, using all the outputs of the classifiers, a unit, called JudgeUnit, determines the cell category (Fig. 3).

In our method, JudgeUnit is constructed by 1) a weighted majority voting method, 2) Random forest[12], 3) FCN. In the weighted majority voting method, the voting of a normal (or abnormal) cell is calculated by the sum of the the possibilities of a normal (or abnormal) cell obtained by the classifiers.

Our cell classification system includes seven classifies: DNN-FC, DNN-LSTM, and five VGG16s. In the training of the classifiers, the dataset is divided into three sets: 1) the set $\mathcal{S}_1$ of the data for training the classifier, 2) the set $\mathcal{S}_2$ of the data for training JudgeUnit, and 3) the set $\mathcal{S}_3$ of the test data for verifying the system. This data distribution method is referred to [13]. All the classifiers are trained by using $\mathcal{S}_1$. After finishing the training of the classifier, the output of the trained classifier using $\mathcal{S}_2$ is used as the input of JudgeUnit.

## 4   Experimental results

To verify the applicability of the proposed method, we made experiments of identifying cells using our dataset with 24,540 cell images described in Sec. 2. In this experiment, we split the training data into 10 sub-datasets, each of which has 2,454 images of 2,400 normal and 54 abnormal cells. Using 10 sub-datasets, 10-fold cross validation is applied to evaluate the proposed system, that is, eight sub-datasets are used as the set $\mathcal{S}_1$ of the data for training the classifier while one sub-dataset is employed as the set $\mathcal{S}_2$ of the data for training JudgeUnit. The rest sub-dataset is the set $\mathcal{S}_3$ of the test data. Moreover, the data augmentation

Table 1: Performances of single classifiers

| Classifiers | Precision | Recall | F-score | Classifiers | Precision | Recall | F-score |
|---|---|---|---|---|---|---|---|
| DNN-LSTM | 0.855 | 0.939 | 0.895 | VGG16($+4.0\mu m$) | 0.957 | **0.961** | **0.959** |
| DNN-FC | 0.890 | 0.954 | 0.921 | VGG16($+6.0\mu m$) | **0.958** | 0.939 | 0.948 |
| VGG16($+0.0\mu m$) | 0.882 | 0.939 | 0.910 | VGG16($+8.0\mu m$) | 0.868 | 0.844 | 0.856 |
| VGG16($+2.0\mu m$) | 0.878 | 0.950 | 0.913 | | | | |

described in Sec. 2 is applied to the two sets $\mathcal{S}_1$ and $\mathcal{S}_2$ to increase the number of the cell images.

Considering the imbalanced test dataset $\mathcal{S}_3$ including 2,400 normal and 54 abnormal cells, the performance of the proposed system is evaluated by four measurements: positive predictive value (PPV), false discovery rate (FDR), negative predictive value (NPV), and false omission rate (FOR). The four measurements are formulated by using true positive (TP), true negative (TN), and false positive (FP), and false negative (FN):

$$PPV = \frac{TP}{TP + FP}; \qquad FDR = \frac{FP}{TP + FP} = 1 - PPV;$$

$$NPV = \frac{TN}{TN + FN}; \qquad FOR = \frac{FN}{TN + FN} = 1 - NPV.$$

Using the four measurements, precision, recall and F-score are defined by

$$Precision = \frac{PPV}{PPV + NPV} \tag{1}$$

$$Recall = \frac{PPV}{PPV + FDR} \tag{2}$$

$$F - score = \frac{2 \cdot Precision \cdot Recall}{Precision + Recall} \tag{3}$$

Firstly, we evaluate the accuracy of the seven classifiers: DNN-FC, DNN-LSTM, and the five VGG16s. (Sec.3.1). Here, VGG16 using the cell images acquired at the focus length of $k$ ($k = 0, 2, 4, 6, 8$) [$\mu m$] is denoted as VGG16($k$).

Table 1 shows the values of precision, recall and F-score obtained by the seven classifiers. From the table, VGG16 is highly efficient than DNN-LSTM and DNN-FC because of the followings. VGG16, DNN-LSTM and DNN-FC have deeper convolutional layers. Since VGG16 is pretrained by ImageNet, VGG16s tend to be trained with acceptable accuracy. On the contrary, the initial weights of DNN-LSTM and DNN-FC are selected randomly. Generally, the initial weights influences on the accuracy of training DNNs. Therefore, the determination of the initial weights of DNN-LSTM and DNN-FC is one of our future works. Among the five VGG16s, VGG16(4) achieves the best performance because many images at the focus length of $+4.0\mu m$ are in focus with less blurred.

Secondly, we construct five types of the cell classification systems composed of the following classifiers:

Table 2: Comparions of the proposed system, System1,2,3 and 4.

| Classifiers | Voting | | | Random forest | | | FCN | | |
|---|---|---|---|---|---|---|---|---|---|
| | Precision | Recall | F-score | Precision | Recall | F-score | Precision | Recall | F-score |
| Proposed | 0.959 | 0.933 | 0.946 | **0.948** | 0.974 | **0.961** | 0.959 | **0.974** | **0.966** |
| System1 | 0.963 | 0.954 | 0.958 | 0.940 | **0.981** | 0.960 | 0.957 | 0.970 | 0.964 |
| System2 | 0.972 | 0.948 | **0.960** | 0.942 | 0.974 | 0.958 | **0.962** | 0.970 | **0.966** |
| System3 | **0.967** | 0.950 | 0.959 | 0.940 | 0.978 | 0.959 | 0.957 | 0.970 | 0.964 |
| System4 | 0.898 | **0.956** | 0.926 | 0.914 | 0.930 | 0.922 | 0.916 | 0.950 | 0.933 |

**Our proposed system :** DNN-LSTM, DNN-FC, and VGG16($k$) ($k = 0, 2, 4, 6, 8$)
**System1 :** DNN-LSTM, DNN-FC, and VGG16($k$) ($k = 0, 4, 8$)
**System2 :** VGG16($k$) ($k = 0, 2, 4, 6, 8$)
**System3 :** VGG16($k$) ($k = 0, 4, 8$)
**System4 :** DNN-LSTM and DNN-FC.

Moreover, the three kinds of JudgeUnits (a weighted majority voting method, Random forest, and FCN) are applied to all the systems. Therefore, we constructed 15 cell classification systems.

Table 2 show the values of precision, recall and F-score obtained by the constructed systems. The combination of the proposed method and FCN achieves the best performance among other systems. Especially, the accuracy of System2 and 3 is higher than that of VGG16(4) which is best among other classifiers. Moreover, compared with System2 and 3 using only VGGs, the proposed system and System1 using different types of the classifiers improve the accuracy of identifying cells. These results shows the effectiveness of combining the different types of classifiers.

For each system, the systems using FCN as JudgeUnit tend to achieve the highest F-score compared with those using Random forest and a weighted majority voting method. This is because FCN can effectively aggregate the feature of multi focus length images. From the experimental results, our proposed system using FCN can classify cells with high performance.

## 5   Conclusion

In this paper, we proposed an automatic system for classifying normal and abnormal cells from multi-focus images by combining multiple DNNs with different architectures. From the experimental results, the combination of the classifiers leads to the accuracy improvement of the classification system.

Our future works include the development of classifiers using multi-focus images and the framework of combining the classifiers to improve the cell classification accuracy.

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
