# OpenReview forum: "Cell Classication from Multi-Focus Images by Deep Neural Network"
_MICCAI.org/2019/Workshop/COMPAY — Submitted to COMPAY 2019_

### Official Review · AnonReviewer3 · 2019-07-26
**Cell classification from multi-focus images by deep neural network.**

**Rating:** 4
**Confidence:** 5

**Review:**

Summary
The authors propose a classification system to discern between normal and abnormal cells. In particular, they feed the classifier with a series of images of the same cell taken at different focus points, mimicking what humans do in the clinical setting. They present a variety of architectures that are able to integrate these images, and compare their performance on a test set.

Strong points:
* The problem statement is clear and sound, addressing a relevant clinical problem.
* They compare a large variety of techniques, using deep learning and non-deep learning based methods.

Constructive criticism:
* Major: the entire evaluation procedure is flawed. They compare the different methods using precision-recall-f1 metrics, however, these metrics are based on a fixed undisclosed detection threshold (operating point). It is incorrect to do so since you can tune the detection threshold to obtain any F1-score. The optimal detection threshold could be 0.5, or 0.8, or 0.2, we don’t know a priori. I recommend using the area under the either ROC curve or F1-score curve to be independent of this threshold. For example, for a given method you compute a series of F1-score measurements by changing the detection threshold between 0 and 1, then compute the area under that curve. The higher the area, the better the classifier (more robust and accurate). Because of this issue, it is not possible to say which method is better.
* Major: I think a fundamental question to answer is whether using a series of multi-focus images is beneficial over using just one focus. I am not convinced by this idea since it seems that all the information about the cell is provided with +4 focus image. In order to answer this question, the authors have to provide a strong baseline using single-focus images. By strong baseline, I mean a similar ensemble system than the one used for multi-focus. It could be that an ensemble of networks (not just one network) trained on single-focus is enough to achieve competitive performance, this is not tested as far as I know.
* Major: given the amount of complexity in the classifier (ensemble of methods), the training and evaluation procedure should be repeated several times, providing a confidence interval or mean+std measurements in order to compare the methods reliably. Otherwise, the difference between methods could be explained by different random initializations.
* Minor: several design choices are not motivated enough. For example, why did the authors trained an FCN on top of probabilities instead of using previously concatenated classifier’s features? It seems very suboptimal for neural network training. This decision forces you to have to train two systems using S1 and S2 sets, instead of one system end-to-end.
* Minor: there are a few spelling mistakes throughout the paper (see the third line in the abstract, or the term “enter sample” in the introduction, for example).
* Minor: given the limited extension of the submission, I would omit the formulas to explain precision, recall and F1-score, since they are standard metrics in machine learning.

---

### Official Review · AnonReviewer1 · 2019-08-02
**Interesting parts but issues with results**

**Rating:** 4
**Confidence:** 3

**Review:**

The authors develop classification methods for cytology screening, utilizing the information of several focus planes at once. The work is well motivated from a clinical usefulness standpoint, and the idea to use several focus planes is interesting to explore. The paper is generally clear and easy to follow, be it with a few language errors.

The authors propose a scheme combining several DNN classifiers. The rationale for the quite complex scheme chosen is a bit vague, but that might be ok in case the results demonstrate strong performance.

Unfortunately there is a mistake in the equations of the performance measures – the derivation of precision and recall are incorrect. Precision = PPV = tp / (tp + fp), and Recall = tp / (tp + fn). If the incorrect equations have been used for the results, they are invalid. I will assume that it is a mere writing error and that the results are correctly described.

The authors conclude that the differences in results demonstrate that the classifier combination improves accuracy. In my view, the results are not clear enough to make such a statement. Only a single operating point is studied, rather than giving an AUC. Even if assuming that the differences will hold across operating points, the results are inconclusive to me. What I take away from table 1 is that the comparison is not really fair due to the pre-training of VGG16. It would have been interesting to delve deeper into whether an approach utilizing the image relationship could be useful. What I take away from table 2 is that the differences are very small, I’m not convinced that it will translate to meaningful performance differences in a real-world scenario. I’d like to see a bit of discussion about the performance numbers given the imbalanced data set. What are reasonable targets when there are so few abnormal cells to find, and what can be considered a substantial quality difference? Furthermore, the difference compared to the baseline of a single VGG16 on the middle focus layer is also limited, which makes me doubt that the extra complexity is worthwhile.

A couple of minor comments is that how the cell images are automatically extracted should be described, and that the bold face in the leftmost numeric column of table 2 appears to be in the wrong place (at System3, should be at System2).

To conclude, I think it is fine to either have a really interesting method proposal, even without competitive results, or competitive results with more vaguely motivated methods. In this case, unfortunately, the paper as it stands now falls slightly short in both these aspects.

---

### Official Review · AnonReviewer2 · 2019-08-08
**Cell Classification from Multi-Focus Images by Deep Neural Network**

**Rating:** 4
**Confidence:** 3

**Review:**

The paper describes a method for classification of cancerous and non-cancerous cell images. The approach is applied to multi-focus images and consists of a combination of multiple networks. The approach is evaluated in a 10-fold cross validation with a large, clinical dataset.

In my view the interesting aspect of this method is the use of multi-focus images. I believe the authors could create a very useful study into the added value of such images. The aim of the paper as they describe it, however, is to improve the performance of classification systems for imbalanced data. This is not demonstrated in the paper. I cannot tell from the results that their combination of networks results in better (more robust and/or more accurate) classification for data with a high class imbalance.

The methodology in the paper is rather complex. A large number of networks and combinations thereof are presented. These have been initialised in different ways, making a good comparison between them even harder. The motivation for these design choices is not always clear.

The classification problem is simplified somewhat from the real-life situation, by decreasing the number of classes. Reading between the lines, I get the impression this was mainly done because the gold standard would otherwise have been less reliable.

Overall, I think the authors would have a stronger paper if they focussed on the multi-focus data rather than the class imbalance. If they could develop good methods for such data and investigate the added value of the information, it would be a very interesting paper. Potentially, the multi-focus data could improve classification of such a tough, full six-class problem.